# Expression of PD-1 and CTLA-4 Are Negative Prognostic Markers in Renal Cell Carcinoma

**DOI:** 10.3390/jcm8050743

**Published:** 2019-05-24

**Authors:** Andreas Kahlmeyer, Christine G. Stöhr, Arndt Hartmann, Peter J. Goebell, Bernd Wullich, Sven Wach, Helge Taubert, Franziska Erlmeier

**Affiliations:** 1Department of Urology and Pediatric Urology, University Hospital Erlangen, Friedrich Alexander-University Erlangen-Nuernberg, 91054 Erlangen, Germany; peter.goebell@uk-erlangen.de (P.J.G.); bernd.wullich@uk-erlangen.de (B.W.); sven.wach@uk-erlangen.de (S.W.); helge.taubert@uk-erlangen.de (H.T.); 2Institute of Pathology, University Hospital Erlangen, Friedrich Alexander-University Erlangen-Nuernberg, 91054 Erlangen, Germany; christine.stoehr@uk-erlangen.de (C.G.S.); arndt.hartmann@uk-erlangen.de (A.H.); f.erlmeier@icloud.com (F.E.); 3Pathology Muenchen-Nord, 80992 Munich, Germany

**Keywords:** CTLA-4, PD-1, PD-L1, renal cell carcinoma, prognostic marker, immunohistochemistry, mortalitiy

## Abstract

Immuno-oncological therapy with checkpoint inhibition (CI) has become a new standard treatment in metastatic renal cell carcinoma (RCC), but the prognostic value of the expression of CI therapy target molecules is still controversial. 342 unselected consecutive RCC tumor samples were analyzed regarding their PD-1, PD-L1, and CTLA-4 expression by immunohistochemistry (IHC). The prognostic values for cancer-specific survival (CSS) and overall survival (OS) were analyzed for those not exposed to CI therapy. The expression of PD-1 in tumor-infiltrating mononuclear cells (TIMC) and PD-L1 in tumor cells was detected in 9.4% and 12.3%, respectively (Immune reactive score (IRS) > 0). Furthermore, PD-L1 expression in TIMC (IRS > 0) and CTLA-4 expression in TIMC (>1% positive cells) was detected in 4.8% and 6.3%. PD-1 expression and CTLA-4 expression were significantly associated with a worse OS and CSS in log rank survival analysis and univariate Cox regression analysis. CTLA-4 expression is a prognostic marker that is independently associated with a worse outcome in multivariate Cox regression analysis in the whole cohort (OS: *p* = 0.013; CSS: *p* = 0.048) as well as in a non-metastatic subgroup analysis (OS: *p* = 0.028; CSS: *p* = 0.022). Patients with combined CTLA-4 expression and PD-1-expression are at highest risk in OS and CSS. In RCC patients, PD-1 expression in TIMC and CTLA-4 expression in TIMC are associated with a worse OS and CSS. The combination of PD-1 expression in TIMC and CTLA-4 expression in TIMC might identify high risk patients. This is, to our knowledge, the first description of CTLA-4 expression to be a prognostic marker in RCC.

## 1. Introduction

With a worldwide incidence of about 338,000 in 2012, kidney cancer accounts for 2–3% of all malignant tumors [1,2]. Approximately 85–95% are renal cell carcinomas (RCC), with an increasing incidence worldwide [3]. 75–80% are clear cell renal cell carcinomas (ccRCC), followed by papillary, chromophobe, and other histological subtypes [4,5]. The five-year overall survival (OS) in localized RCC is approximately 90%, but the median OS in metastatic diseases is only about 18–22 months [6,7,8]. Renal cell carcinoma is considered to be an immunogenic tumor [9]. The characterization of immune subtypes has revealed that renal cell carcinoma belongs mainly to the inflammatory subtype with an increased leukocyte fraction [10], and enhanced MHC-I expression is a good prognostic factor in ccRCC [11]. Accordingly, current checkpoint inhibition (CI) therapies show promising effects in RCC patients by inhibiting two of the immune escape mechanisms addressing the differentiation and activity of effector T cells [12].

The most targeted immune escape mechanism is the down-regulation of T cell activity by a programmed cell death protein 1/programmed cell death ligand 1 (PD-1/PD-L1) interaction in the tumor microenvironment. Tumor-infiltrating mononuclear cells (TIMC) in RCC show an increased expression of PD-1 in comparison to lymphocytes in peripheral blood. This is considered to be a marker for limited cytotoxic activity in the tumor [13,14], but retrospective analyses to the prognostic value of PD-1 expression in RCC are contradictory [15,16].

The expression of PD-L1 on tumor cells in RCC is associated with a higher tumor stage, a worse response to tyrosine kinase inhibitor (TKI) therapy, and a worse prognosis [17,18,19,20,21,22,23]. The expression of PD-L1 in TIMC in RCC is associated with a worse prognosis in ccRCC [17], but the relevance of PD-L1 expression in TIMC in other histological subtypes remains unknown [19].

The second frequently targeted pathway is the regulation of the initial priming of naive T cells in lymph nodes via cytotoxic T-lymphocyte-associated Protein 4/B7 protein (CTLA4/B7) signaling [24,25,26]. The interaction of CTLA-4, expressed by activated T cells and regulatory T cells [27], with B7-1 and B7-2 on T cells can limit or decrease their activation [28,29] and downregulate proliferation and interleukin-2 (IL-2) secretion [24]. In RCC, about 1% of TIMC express CTLA-4, and expression increases with higher tumor stages. In papillary RCC, up to 2·7% of TIMC express CTLA-4 [30]. Polymorphisms in the CTLA-4 gene are associated with a higher risk for high-stage ccRCC [31] and are associated with better OS in metastatic patients treated with TKI [32].

The aim of the study was to assess the prognostic value of the expression of checkpoint inhibitor targets in a population not treated with CI. The published prognostic significance of the expression of CI therapy targets in populations under checkpoint inhibition therapy [33] can also be better assessed.

## 2. Experimental Section

Consecutive and unselected tumor specimens from 453 patients undergoing radical or partial nephrectomy for RCC between 1998 and 2011 at the Department of Urology and Pediatric Urology at the University Hospital Erlangen were collected. No tissue samples of renal metastases from renal cell carcinomas or from carcinomas of other origin were included. Tumors already metastasized at the time of surgery are referred to as primary metastatic diseases; tumors metastasized during follow-up are referred to as secondary metastatic diseases. Tumors have been reevaluated independently by two experienced pathologists (AH, FE), and the histological subtype was reclassified according to the UICC 2010 TNM tumor staging system. Construction of the tissue microarray (TMA) has been described in detail previously [34]. All patients, beginning from 2008, gave informed consent. For samples before the 2008 Ethic Commission in Erlangen, all patients waived the need for informed individual consent. The study is based on the approvals of the Ethic Commissions of the University Hospital Erlangen (No.3755). The study was carried out according to the latest version of the Declaration of Helsinki and is approved by the institutional ethics committee.

The expression of PD-1, PD-L1, CD3, and CTLA-4 was investigated by immunohistochemistry (IHC) on 3 µm sections from formalin-fixed paraffin-embedded TMA tissue blocks. For CTLA4-staining, we used a mouse monoclonal anti-CTLA4/CD152 antibody (clone BSB-88, BSB2883, dilution 1:50, BioSB, Santa Barbara, CA, USA). The CTLA4/CD152 antibody was validated on a multi tissue TMA containing colon, breast, small intestine, adrenal, uterus, prostate, ovary, liver, tonsil, salivary gland, brain, heart, renal, appendix, skin, nerve, lung, testis, placenta, spleen, pancreas, endometrium, stomach, and parotid tissue. The CTLA4/CD152 antibody demonstrated specific staining in lymphatic tissue. Therefore, lymphatic tissue in a normal tonsil was used as a positive control. Upon evaluation of the CTLA4-stained TMAs, negative control slides without the addition of the primary antibody were added as internal negative controls. For enhancement and visualization, we used an EnVision + System, HRP (Dako, Agilent Technologies GmbH & CoKG, Hamburg, Germany) according to the manufacturer’s instructions. PD1, PD-L1, and CD3 stainings were performed on a Ventana Benchmark ULTRA staining system using standard protocols and a mouse monoclonal anti-PD1 antibody (clone NAT105, Ventana 760-4895, ready to use, Cell Marque^TM^, Merck KGaA, Darmstadt, Germany), a rabbit monoclonal anti-PD-L1antibody (clone28-8, ab205921, 1:200, Abcam plc, Cambridge, UK), and a rabbit monoclonal anti-CD3-antibody (clone SP7, RBG024, 1:150, Zytomed-Systems GmbH, Berlin, Germany), respectively. The secondary reaction was performed using a Ventana ultraView Universal DAB Detection Kit (Ventana Medical Systems, Tucson, AZ, USA) for anti-PD1 and anti-CD3 and a Ventana OptiView DAB IHC Detection Kit (Ventana Medical Systems, Tucson, AZ, USA) for anti PD-L1. All procedures were performed according to the manufacturer’s instructions. We used hematoxylin for counterstaining. IHC-staining was semi quantitatively assessed using the immunoreactivity score (IRS) [35]. The intensity of the staining was classified in 4 categories (no staining (0), weak staining (1), moderate staining (2), and strong staining (3)) and multiplied by the categorized proportion of positive cells (no (0), <10% (1), 10–50% (2), 51–80% (3) and >80% (4)). An IRS = 0 was considered as negative, and an IRS > 0 was counted as positive. PD1 staining was analyzed on TIMC. PD-L1 staining on tumor cells and on TIMC was assessed separately. As no cut-off value for CLTA-4 expression in IHC in RCC is defined, and CTLA-4 expression has been described in about 1% of TIMC in RCC [30], we defined a cut off at ≥2% CTLA-4 positive cells in tumor tissue infiltrating TIMC for defining a tumor as CTLA-4 positive. CD3 positive cells were counted in 4 high power fields, and the number of CD3 positive cells per high power field was recorded. Thirty additional controls revealed no PD1-, PD-L1, or CTLA-4 staining in corresponding normal renal tissue distant to RCC. All stained TMAs were assessed independently by two experienced pathologists (AH, FE), both blinded for clinical data. In cases for which the results were inconsistent, the pathologists worked to reach a consensus.

After the exclusion of missing tissue, missing tumors, and incomplete survival data, 342 tumor-representative specimens could be assessed. There were no differences in the clinical and histopathological features between evaluable cases and the entire tumor cohort. Statistical analyses was performed using IBM SPSS Statistics 21 (IBM-Corporation Germany GmbH, Ehningen, Germany). The nonparametric correlation was assessed by the two-sided Spearman Rho test. Survival analysis was done with the log-rank test, and Kaplan-Meier survival curves were drawn. The association of marker expression and survival was assessed by univariate and multivariate Cox regression analysis. Differences were regarded statistically significant at *p* < 0.05.

Individual participant data that underlie the results reported in this article will be shared after de-identification in the Appendix A. Data will be available immediately following publication for 5 years.

## 3. Results

### 3.1. Patients’ Characteristics and Expression of Target Molecules of CI Therapies

Three hundred forty-two tumor specimens were analyzed, with 64.6% (221) from male patients (Table 1). The median age at surgery was 66 years (23–92 years). 10.8% (37) of the patients presented with primary metastatic diseases, and 12.6% (43) developed secondary metastases within a median follow up period of 38 months (1–160 months). A histopathological evaluation showed that 78.9% (270) were clear cell RCC, 12.0% (41) were papillary RCC, 7.0% (24) were chromophobe RCC, and 2% (7) were other histopathological subtypes (five hybrid mixed tumors, two sarcomatoid dedifferentiated tumors) (Table 1). The expression of PD-1 in TIMC, PD-L1 in tumor cells, PD-L1 in TIMC, and CTLA-4 in TIMC was detected in 9.4% (31), 12.3% (41), 4.8% (16), and 6.3% (20), respectively (Figure 1, Table 1). PD-1 expression in TIMC is associated with a high grade tumor (G3, *p* < 0.001, correlation coefficient 0.215) or primary metastatic diseases (*p* = 0.007, not significant with Bonferroni correction, correlation coefficient 0.149). PD-L1 staining in tumor cells is associated with the papillary subtype (*p* < 0.001, correlation coefficient 0.242), a high grade tumor (G3, *p* = 0.029, not significant with Bonferroni correction, correlation coefficient 0.120), or secondary metastatic diseases (*p* = 0.030, not significant with Bonferroni correction, correlation coefficient 0.125). The CTLA-4 expression in TIMC is associated with primary metastatic diseases (*p* = 0.006, not significant with Bonferroni correction correlation coefficient 0.153). The simultaneous expression of PD-1 and CTLA-4 is associated with a higher tumor stage (*p* ≥ T3, *p* = 0.037, not significant with Bonferroni correction, correlation coefficient 0.119) and a high grade tumor (G3, *p* = 0.005, not significant with Bonferroni correction, correlation coefficient 0.159), as well as with primary metastatic diseases (*p* < 0.001, correlation coefficient 0.200). Tumor immune infiltration assessed by CD3 rate is higher in male patients (*p* = 0.026, not significant with Bonferroni correction, correlation coefficient 0.122), ccRCC (*p* = 0.035, not significant with Bonferroni correction, correlation coefficient 0.122), or high grade diseases (G3, *p* = 0.013, not significant with Bonferroni correction, correlation coefficient 0.137) (Table 1). A nonparametric correlation revealed a significant association between PD-1, PD-L1, and CTLA-4 expressions (Table 2).

### 3.2. Survival Analysis

#### 3.2.1. Log Rank Test

A survival analysis showed a significantly longer estimated OS in patients with PD-1 negative TIMC (Table 3). The estimated mean OS advantage was 109.7 vs. 55.8 months (*p* = 0.002) (Figure 2). A similar trend was detected in estimated mean cancer specific survival (CSS), although this did not reach statistical significance (142.1 vs. 84.5 months, *p* = 0.072) (Figure 3). The positive CTLA-4 expression in TIMC shows a significant association with a poor estimated mean OS (84.1 vs. 107.76 months, *p* = 0.013) (Figure 4) and CSS (125.5 vs. 140.6 months, *p* = 0.019) (Figure 5). Especially within the first year after a resection, CTLA-4 positive patients performed worse. A small subgroup of patients with a positive PD-1 expression in TIMC and a positive CTLA-4 expression in TIMC are at high risk in their estimated mean OS (29.8 vs. 108.8 months, *p* < 0.001) (Figure 6) and CSS (39.3 vs. 142.4 months, *p* < 0.001) (Figure 7).

A subgroup analysis with clear-cell histology tumors only shows comparable results regarding the prognostic value of PD-1 and CTLA-4 expression (Table 4). Though PD-1 expression in TIMC is not associated with an estimated mean OS (65.9 vs. 116.2 months, *p* = 0.058) and CSS (92.4 vs. 148.0 months, *p* = 0.329) in primary non-metastatic patients, CTLA-4 expression in TIMC (OS 94.9 vs. 114.8 months, *p* = 0.041; CSS 125.3 vs. 146.8 months, *p* = 0.001), and the combination of PD-1 expression in TIMC and CTLA-4 expression in TIMC are significantly associated with a worse estimated mean OS (32.0 vs. 116.2 months, *p* = 0.001) and CSS (32.0 vs. 148.7 months, *p* = 0.001) (Table 5). For PD-L1-expression in tumor cells and TIMC, as well as for infiltration by CD3 positive cells, no association with an estimated mean OS or CSS was found.

#### 3.2.2. Univariate Cox Regression Analysis

A univariate Cox regression analysis shows a significant prognostic value with regard to OS and CCS for a high tumor stage (≥pT3; OS: HR = 3.4, *p* < 0.001; CSS: HR = 3.0, *p* = 0.001), a high tumor grade ( = G3; OS: HR = 3.5, *p* < 0.001; CSS: HR = 6.7, *p* < 0.001), and an advanced age at diagnosis (>65 years; OS: HR = 2.7, *p* < 0.001; CSS: HR = 2.3, *p* = 0.023). The male gender (OS: HR = 1.8, *p* = 0.016) and a higher eastern cooperative oncology group (ECOG) performance status (ECOG > 0; OS: HR = 2.3, *p* < 0.001) were associated with a worse OS but not with CSS. As shown in the log rank test, PD-1 expression in TIMC (HR = 2.5, *p* = 0.003) and CTLA-4 expression in TIMC (HR = 2.4, *p* = 0.017) are significantly associated with a worse OS. CTLA-4 expression in TIMC is the only prognostic IHC marker for CSS (HR = 3.3, *p* = 0.027). The PD-1 and CTLA-4 double positive subgroup is again associated with a worse OS (HR = 4.3, *p* = 0.002) and a worse CSS (HR = 6.3, *p* = 0.003) (Appendix A). In primary non metastatic patients, only a high tumor grade (G3, HR = 4.4, *p* = 0.001), CTLA-4 expression (HR = 2.5, *p* = 0.005), and the combination of PD-1 and CTLA-4 expression (HR = 16.3, *p* < 0.001) are significantly associated with a worse CSS (Appendix A).

#### 3.2.3. Multivariate Cox Regression Analysis

After controlling for the univariately significant parameters age, gender, tumor grade, tumor stage, and ECOG performance status, CTLA-4 expression in TIMC remains independently prognostic for OS (HR = 2.8, *p* = 0.013) and CSS (HR = 3.7, *p* = 0.048) in the whole cohort (Table 6), as well as in the ccRCC-only subgroup analysis (HR = 4.1, OS *p* = 0.006; CSS HR = 8.2, *p* = 0.003) (Table 7) and in the primary non-metastatic subgroup analysis (OS HR = 3.4, *p* = 0.028; CSS HR = 7.4, *p* = 0.022) (Table 8). PD-1 expression in TIMC alone and in combination with CTLA-4 expression in TIMC is not an independent prognostic factor in multivariate Cox regression analysis.

## 4. Discussion

As part of the anti-tumor immune response, naive T cells are activated by the presentation of tumor antigens from dendritic cells and other antigen-presenting cells via Major Histocompatibility Complex (MHC) molecules. In this immunological checkpoint, costimulatory signals for the activation or anergy of the effector T cell are crucial. Two of these immunosuppressive costimulatory pathways are the PD-1/PD-L1 interaction and the activation of CTLA-4 via B7-1 or B7-2. Recently, the combination of checkpoint inhibitors ipilimumab (anti-CTLA-4 antibody) and nivolumab (anti-PD-1 antibody) in the phase III Checkmate 214 study (NCT02210117) in advanced RCC demonstrated the statistically significant improvement of overall response rate (ORR) compared to the standard of care with sunitinib in first line therapy in intermediate and poor risk patients [36]. Though checkpoint inhibitors targeting these pathways are changing therapy in metastatic renal cell carcinoma [12,36], the prognostic value of PD-1, PD-L1, and CTLA-4 expression in localized renal cell carcinoma remains unclear.

PD-1 on the cell surface of activated T cells is immunosuppressive when it is activated in peripheral tissue by tumor cells via PD-L1 or PD-L2 expression. In our analysis, PD-1 expression was detected in TIMC in about 9.4% of our tumor specimens, with significantly higher expression in high grade tumors and primary metastatic diseases. Previous studies on the prognostic value of PD-1 expression in TIMC in ccRCC are contradictory [15,16,37]. In our cohort, we found a significant association with OS and CSS by univariate Cox regression analysis, but, after an adjustment to tumor stage, tumor grade, gender, age, and ECOG performance status, multivariate Cox regression analysis revealed that it is not an independent prognostic factor.

PD-1 expression on TIMC and PD-L1 expression in tumor cells are considered to be markers of reduced T cell function in the tumor microenvironment [14]. An association of PD-L1 expression and a poor prognosis is described in ccRCC as well as in other histological subtypes [17,18,20,21,23,37], and recently published data show an association with a worse OS and progression free survival (PFS) in high risk non metastatic diseases [38]. In our analysis, PD-L1 expression is significantly correlated with high grade diseases and non-metastatic diseases and lower in secondary metastatic diseases. However, no association with OS or CSS could be detected.

CTLA-4 is the second target of checkpoint inhibition therapies in renal cell carcinoma. Antibodies to CTLA-4 were the first checkpoint inhibitors with anti-tumor activity [39,40]. CTLA-4 expression was significantly correlated with primary metastatic diseases and associated with a a reduced OS and CSS in the whole cohort, as well as in a ccRCC-only subgroup. Though associated with primary metastatic diseases, CTLA-4 expression in a non-metastatic disease subgroup is still significantly associated with a worse OS and CSS. Multivariate analysis revealed CTLA-4 expression as an independent prognostic factor after an adjustment to tumor grade, tumor stage, age, gender, and ECOG performance status. This is, to our knowledge, the first description of CTLA-4 expression to be a prognostic marker in RCC.

As there is significant cross correlation, the synchronous expression of PD-1 in TIMC and of CTLA-4 in TIMC identifies patients with a worse outcome. The estimated median OS is more than six years shorter in this subgroup (29.8 vs. 108.8 months, *p* = 0.001) in the whole cohort and more than seven years shorter (32.0 vs. 116.2 months, *p* < 0.001) in the primary non-metastatic subgroup. The simultaneous expression of PD-1 in TIMC and CTLA-4 in TIMC seems to be a predictor for rapid disease progression and death, even in localized diseases. As we showed that the combination of PD-1 and CTLA-4 expression is correlated to metastatic diseases, we hypothesize that the expression of these checkpoint molecules may be an early marker for micro metastatic diseases undetectable by cross-sectional imaging. Local therapy alone therefore is not sufficient for long term tumor control, and adjuvant therapy may be considered.

In metastatic diseases, PD-L1 expression in tumor cells or TIMC is the most studied biomarker for the prediction of a response to PD-1/PD-L1 CI therapy [33,41,42]. Response rates are better in PD-L1 positive tumors, but, as there are relevant response rates in PD-L1 negative subgroups, PD-L1 expression still is regarded as a prognostic but not predictive marker and, therefore, cannot be recommended for therapy allocation [43,44]. Major problems in the development of predictive biomarkers for CI therapy are the dynamic expression, the heterogeneity within the primary tumor, and the low correlation between the primary and metastatic sites [45,46,47]. In primary non-metastatic diseases, the whole tumor could be assessed after a resection directly prior to allocation to adjuvant therapy. The expression of key molecules for immune escape mechanisms could therefore be more representative for the immune status of the whole disease. In our view, inhibition of tumor specific immune escape mechanisms in high risk patients at a non-metastatic stage could become an important part of RCC therapy. Large scale trials on adjuvant CI therapies in RCC with antibodies targeting PD-1 and CTLA-4 (NCT03138512; NCT03142334; NCT03288532; NCT03024996) are initiated or ongoing. These studies contain large biomarker arms and may reveal predictive markers in the near future [33]. Exploratory analyses in metastatic renal cell carcinoma already show that gene expression analyses allow the classification into new subgroups which may have an association to responses across different treatments [48].

In reflection of an analysis by Choueri et al., we have evaluated the influence of the localization (left vs. right), but neither OS (*p* = 0.165) nor CSS (*p* = 0.10) demonstrated a significant correlation [49]. Thus, we did not consider localization as a separate factor in our multivariate analysis. The limitations of our study are the restriction to IHC expression analysis and the non-interventional retrospective design. There is no reliable information on treatment after a tumor resection, which may bias survival analysis. We were able to show that CTLA-4 expression is an independent marker for high risk of early cancer specific death after local therapy, but evidence for efficiency of CTLA-4 targeting therapy in an adjuvant setting is still missing. Furthermore, our analysis does not contain a comprehensive assessment of the tumor immune status, and the regulation of anti-tumor cytotoxicity in RCC is much more heterogeneous and diverse [50,51]. It would also be desirable to validate the results of our study in an independent cohort.

Another limitation is the relatively small number of patients with non-clear cell histology. Though PD-L1 expression is associated with papillary RCC, no correlation to OS or CSS was detected. For chromophobe RCC, no association of PD-1 and PD-L1 expression with OS was reported previously in a different chromophobe RCC-only cohort by Erlmeier and colleagues [52].

In conclusion, to our knowledge, this is the first description of CTLA-4 expression as an independent prognostic marker for OS and CSS in RCC. It is of special note that the combination of PD-1 and CTLA-4 expression identifies high risk patients with poor OS and CSS.

## Figures and Tables

**Figure 1 jcm-08-00743-f001:**
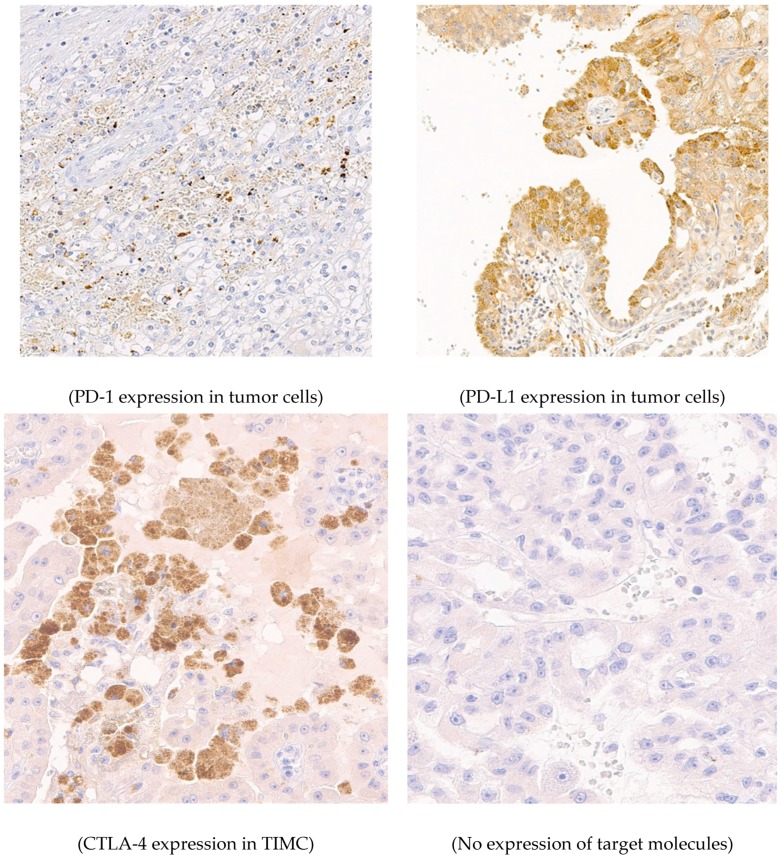
Distribution and morphology of PD-1, PD-L1, and CTLA-4 staining.

**Figure 2 jcm-08-00743-f002:**
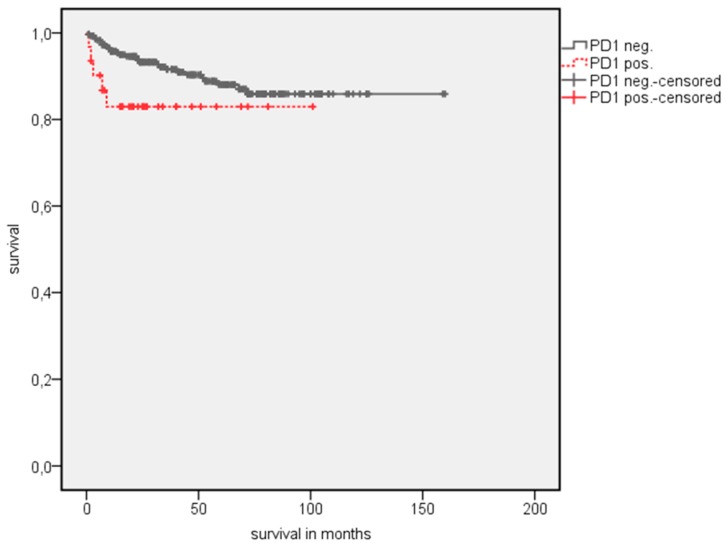
Kaplan Meier analysis: Association of PD1 expression in tumor-infiltrating mononuclear cells (TIMC) with overall survival (OS) in all RCC patients.

**Figure 3 jcm-08-00743-f003:**
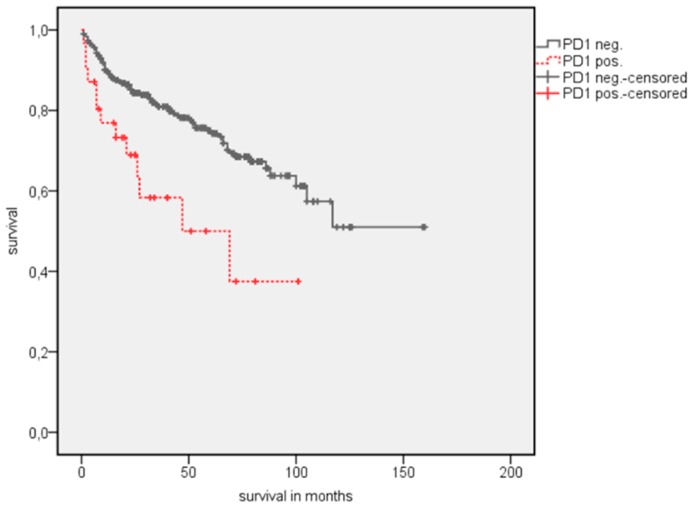
Kaplan Meier analysis: Association of PD1 expression in TIMC with cancer-specific survival (CSS) in all RCC patients.

**Figure 4 jcm-08-00743-f004:**
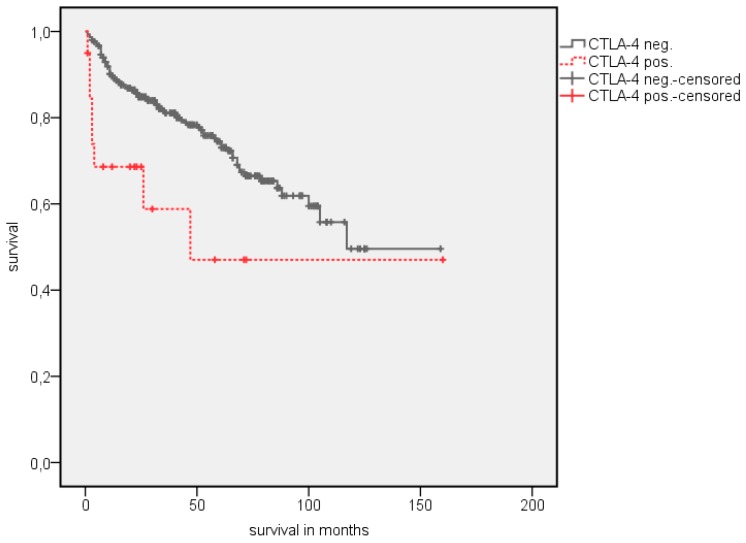
Kaplan Meier analysis: Association of CTLA-4 expression in TIMC with OS in all RCC patients.

**Figure 5 jcm-08-00743-f005:**
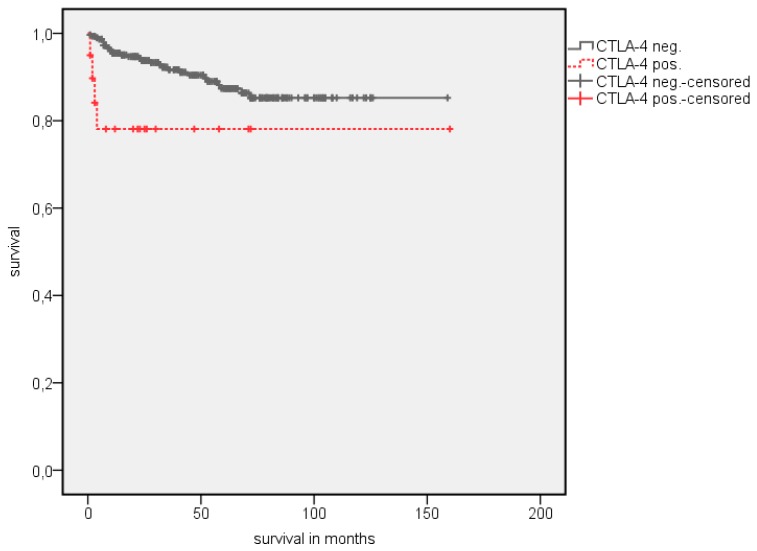
Kaplan Meier analysis: Association of CTLA-4 expression in TIMC with CSS in all RCC patients.

**Figure 6 jcm-08-00743-f006:**
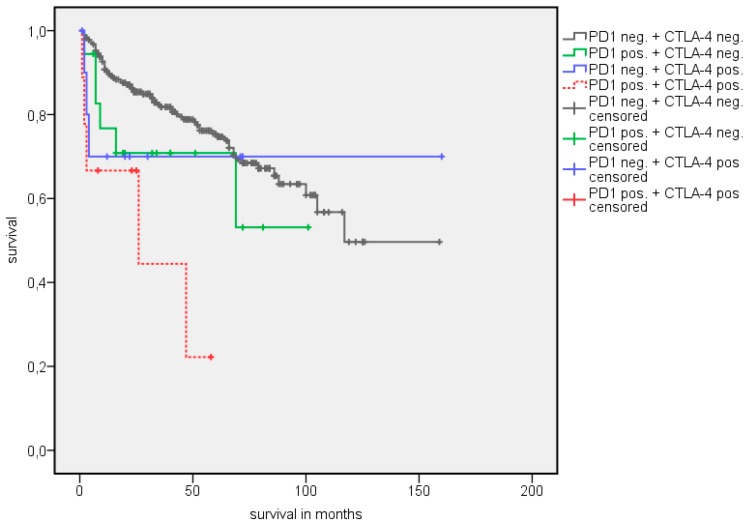
Kaplan Meier analysis: Association of combined PD-1 and CTLA-4 expression with OS in all RCC patients.

**Figure 7 jcm-08-00743-f007:**
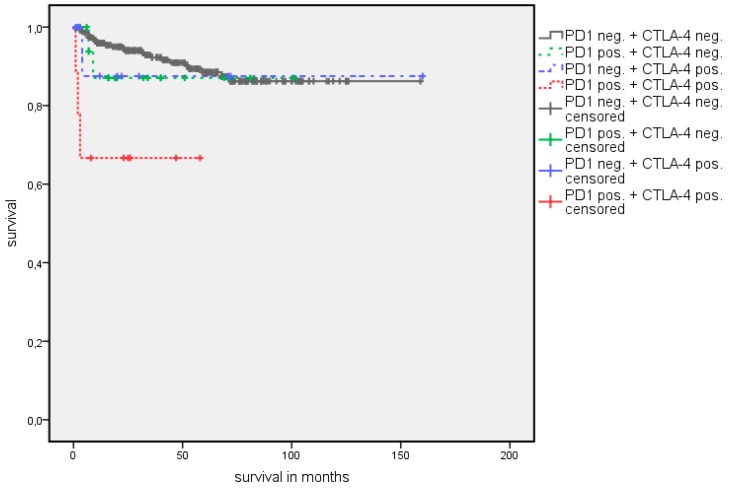
Kaplan Meier analysis: Association of combined PD-1 and CTLA-4 expression with CSS in all RCC patients.

**Table 1 jcm-08-00743-t001:** Patient characteristics and protein expression of immunological markers.

	All RCC	PD-1 Positive in TIMC		PD-L1 Positive in Tumor Cells		PD-L1 Positive in TIMC		CTLA-4 ≥ 2% in TIMC		PD-1 in TIMC Positive and CTLA-4 in TIMC ≥ 2%	
	*n* = 342	*n* = 31 (9.4%)	*p*-Value	*n* = 41 (12.3%)	*p*-Value	*n* = 16 (4.8%)	*p*-Value	*n* = 20 (6.3%)	*p*-Value	*n* = 9 (2.9%)	*p*-Value
Age median	66.0	69		67.0		67.5		69.5		73	
(range)	(23–92)	(40–84)	0.576	(28–83)	0.842	(46–80)	0.887	(46–79)	0.191	(57–79)	0.339
Gender											
female	121 (35.4%)	7 (5.9%)		11 (9.2%)		8 (6.7%)		4 (3.6%)		4 (3.6%)	
male	221 (64.6%)	24 (11.3%)	0.108	30 (14%)	0.210	8 (3.7%)	0.224	16 (7.7%)	0.147	5 (2.5%)	0.064
Histology											
clear cell	270 (78.9%)	25 (9.6%)		24 (9.1%)		10 (3.8%)		15 (6.0%)		7 (2.8%)	
papillary	41 (12.0%)	5 (12.8%)	0.536	13 (32.5%)	<0.001 **	3 (7.7%)	0.265	3 (7.9%)	0.650	2 (5.4%)	0.037
chromophobe	24 (7.0%)	0 (0%)	0.112	1 (4.2%)	0.411	1 (4.2%)	0.929	0 (0%)	0.220	0 (0%)	0.095
other	7 (2%)	1 (14.3%)	0.682	3 (42.9%)	0.003 *	2 (28.6%)	0.002 *	2 (28.6%)	0.017 *	0 (0%)	0.088
Grade											
G1	43 (12.6%)	2 (5.0%)		3 (7.5%)		1 (2.5%)		2 (5.1%)		2 (5.1%)	
G2	219 (64.2%)	13 (6.1%)	0.782	23 (10.6%)	0.547	9 (4.2%)	0.616	12 (5.9%)	0.859	4 (2.0%)	0.862
G3	79 (23.2%)	16 (20.8%)	<0.001 **	15 (19.5%)	0.029*	6 (7.8%)	0.166	6 (8.0%)	0.481	3 (4.1%)	0.005 *
Stage											
pT1 and pT2	247 (74.6%)	18 (7.6%)		29 (12.1%)		12 (5.0%)		12 (5.2%)		4 (1.8%)	
pT3 and pT4	84 (25.4%)	12 (14.3%)	0.071	10 (11.9%)	0.956	4 (4.8%)	0.942	8 (10.0%)	0.135	5 (6.3%)	0.037 *
Metastases											
non-metastatic	262 (76.6%)	18 (7.2%)		37 (14.4%)		15 (5.8%)		12 (4.9%)		5 (2.1%)	
primary metastatic	37 (10.8%)	8 (21.6%)	0.007*	3 (8.1%)	0.482	0 (0%)	0.161	6 (16.7%)	0.006 **	3 (8.3%)	<0.001 **
secondary metastatic	43 (12.6%)	5 (11.6%)	0.320	1 (2.4%)	0.030 *	1 (2.4%)	0.372	2 (4.9%)	0.987	1 (2.4%)	0.620
ECOG											
0	209 (73.6%)	17 (8.3%)		28 (13.7%)		11 (5.4%)		11 (5.7%)		3 (1.6%)	
>0	75 (26.4%)	10 (13.9%)	0.174	9 (12.2%)	0.736	4 (5.4%)	0.997	6 (8.3%)	0.432	4 (5.7%)	0.186
Survival											
OS	250 (73.1%)	18 (58.1%)	0.044 *	29 (70.7%)	0.686	12 (75%)	0.895	12 (60%)	0.173	4 (44%)	0.050 *
CSS	307 (89.8%)	26 (83.9%)	0.233	37 (90.2%)	0.924	16 (100%)	0.175	16 (80%)	0.142	6 (66%)	0.055

* Correlation significant at 0.05 level (2-tailed); ** correlation significant with Bonferroni corrections al 0.001 level (2-tailed).

**Table 2 jcm-08-00743-t002:** Non-parametric correlation of PD-1, PD-L1, and CTLA-4 expression.

		PD-1 Positive in TIMC	PD-L1 Positive in Tumor Cells	PD-L1 Positive in TIMC	CTLA-4 ≥2 % in TIMC	PD-1 in TIMC Positive and CTLA-4 in TIMC ≥ 2%
Pearson Correlation	PD-1 positive in TIMC	1	0.171 **	0.030	0.339 **	0.845 **
Significance (*p*-value)			0.002 **	0.588	<0.001 **	<0.001 **
Pearson Correlation	PD-L1 positive in tumor cells	0.171 **	1	0.044	0.281 **	0.273 **
Significance (*p*-value)		0.002 **		0.423	<0.001 **	<0.001 **
Pearson Correlation	PD-L1 positive in TIMC	0.030	0.044	1	0.131 *	0.106
Significance (*p*-value)		0.588	0.423		0.020 *	0.063
Pearson Correlation	CTLA-4 ≥ 2% in TIMC	0.339 **	0.281 **	0.131 *	1	0.789 **
Significance (*p*-value)		<0.001 **	<0.001 **	0.020 *		<0.001 **
Pearson Correlation	PD-1 in TIMC positive and CTLA-4 in TIMC ≥ 2%	0.845 **	0.273 **	0.106	0.789 **	1
Significance (*p*-value)		<0.001 **	<0.001 **	0.063	<0.001 **	

* Correlation is significant at 0.05 level (2-tailed); ** correlation significant with Bonferroni corrections at 0.002 level (2-tailed).

**Table 3 jcm-08-00743-t003:** Survival analysis (Log rank test) in all renal cell carcinoma (RCC) patients.

All RCC	Estimated Mean OS (months)			Estimated Mean CSS (months)		
	neg.	pos.	*p*-Value	neg.	pos.	*p*-Value
PD-1 TIMC	109.707	55.753	0.002 **	142.071	84.576	0.072
PD-L1 TU	108.923	86.661	0.694	140.353	105.148	0.964
PD-L1 TIMC	106.277	119.932	0.649	No cancer specific death		
CTLA-4	107.758	84.142	0.013 *	140.590	125.534	0.019 *
PD-1 + CTLA-4	108.779	29.778	0.001 **	142.388	39.333	0.001 **
CD3	103.920	112.413	0.628	142.116	138.045	0.390

** Correlation is significant at 0.01 level (2-tailed); * correlation is significant at 0.05 level (2-tailed). neg. negative; pos. positive.

**Table 4 jcm-08-00743-t004:** Survival analysis (Log rank test) in clear cell renal cell carcinomas (ccRCC) patients.

ccRCC	Estimated Mean OS (months)			Estimated Mean CSS (months)		
	neg.	pos.	*p*-Value	neg.	pos.	*p*-Value
PD-1 TIMC	108.796	56.304	0.009 **	139.561	84.690	0.180
PD-L1 TU	108.894	86.841	0.691	138.745	102.306	0.883
PD-L1 TIMC	105.603	121.350	0.484	No cancer specific death		
CTLA-4	107.005	86.924	0.020 *	138.789	113.542	0.004 **
PD-1 + CTLA-4	107.778	24.857	<0.001 **	139.737	34.000	<0.001 **
CD3	102.642	111.860	0.602	138.746	137.061	0.712

** Correlation is significant at 0.01 level (2-tailed); * correlation is significant at 0.05 level (2-tailed).

**Table 5 jcm-08-00743-t005:** Survival analysis (Log rank test) in the no primary metastases subgroup.

No Primary Metastases	Estimated Mean OS (months)			Estimated Mean CSS (months)		
	neg.	pos.	*p*-Value	neg.	pos.	*p*-Value
PD-1 TIMC	116.190	65.937	0.058	147.971	92.391	0.329
PD-L1 TU	115.848	92.812	0.737	146.010	110.674	0.738
PD-L1 TIMC	113.784	119.932	1.000	No cancer specific death		
CTLA-4	114.781	94.893	0.041 *	146.821	125.273	0.001 **
PD-1 + CTLA-4	116.216	32.000	<0.001 **	148.705	32.000	<0.001 **
CD3	108.641	124.215	0.165	146.946	146.186	0.979

** Correlation is significant at 0.01 level (2-tailed); * correlation is significant at 0.05 level (2-tailed).

**Table 6 jcm-08-00743-t006:** Multivariate Cox regression analysis: Association of clinical parameters and CTLA-4 expression with OS or CSS in all RCC patients.

All RCC	OS			CSS		
	Hazard Ratio	Range	*p*-Value	Hazard Ratio	Range	*p*-Value
CTLA-4	2.838	1.248–6.451	0.013 *	3.726	1.011–13.727	0.048 *
Age > 65 years	2.501	1.436–4.356	0.001 **	1.818	0.736–4.487	0.195
Gender male	1.336	0.764–2.335	0.309	0.797	0.333–1.912	0.612
Stage > pT2	3.450	2.119–5.617	<0.001 **	2.520	1.096–5.791	0.030 *
Grade = G3	2.828	1.724–4.641	<0.001 **	9.587	3.689–24.931	<0.001 **
ECOG > 0	1.576	0.952–2.608	0.077	0.477	2.701–1.135	0.775

** Correlation is significant at 0.01 level (2-tailed); * correlation is significant at 0.05 level (2-tailed).

**Table 7 jcm-08-00743-t007:** Multivariate Cox regression analysis: Association of clinical parameters and CTLA-4 expression with OS or CSS in the ccRCC patients.

ccRCC	OS			CSS		
	Hazard Ratio	Range	*p*-Value	Hazard Ratio	Range	*p*-Value
CTLA-4	4.059	1.506–10.940	0.006 **	8.161	2.003–33.26	0.003 *
Age > 65 years	2.132	1.182–3.846	0.012 *	2.162	0.843–5.546	0.109
Gender male	1.220	0.662–2.249	0.524	0.753	0.313–1.81	0.526
Stage > pT2	3.026	1.786–5.301	<0.001 **	1.568	0.623–3.944	0.339
Grade = G3	3.132	1.796–5.462	<0.001 **	11.341	4.108–30.676	<0.001 **
ECOG > 0	1.630	0.928–2.864	0.089 *	1.545	0.612–3.899	0.357

** Correlation is significant at 0.01 level (2-tailed); * correlation is significant at 0.05 level (2-tailed).

**Table 8 jcm-08-00743-t008:** Multivariate Cox regression analysis: Association of clinical parameters and CTLA-4 expression with OS or CSS in the no primary metastases subgroup.

No Primary Metastases	OS			CSS		
	Hazard Ratio	Range	*p*-Value	Hazard Ratio	Range	*p*-Value
CTLA-4	3.370	1.144–9.924	0.028 *	7.351	1.337–40.42	0.022 *
Age > 65 years	2.643	1.395–5.008	0.003 **	1.600	0.437–5.861	0.478
Gender male	1.794	0.906–3.554	0.094	0.983	0.236–4.104	0.981
Stage > pT2	4.115	2.247–7.537	<0.001 **	2.457	0.681–8.868	0.170
Grade = G3	1.605	0.854–3.015	0.142	6.272	1.71–23.01	0.006 **
ECOG > 0	2.071	1.135–3.784	0.018 *	1.261	0.34–4.673	0.729

** Correlation is significant at 0.01 level (2-tailed); * correlation is significant at 0.05 level (2-tailed).

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
