# Peer review of "Expression of PD-1 and CTLA-4 Are Negative Prognostic Markers in Renal Cell Carcinoma"

_jcm, 2019, doi:10.3390/jcm8050743_

Reviewer 1 Report

The authors are to be commended on performing a very large study and examining a very important and novel question. The answers to these questions ( prognostic role of ICI in non CI treated/localized disease patients) , would be very informative in the design and conduct of potential adjuvant studies etc.

However, there are several issues that must be addressed. These include grammar to addressing/clarifying methodology. Some of the suggestions are enumerated below.

Line 16 , CSS , OS was analyzed "in a population not exposed to Immune checkpoint therapy" ( should be added )

Line 38 , should attempt to get more updated data than Heng et al's updat eform 2008 , VEGF TKIs and combination therapy is extending survival.

Line 100 -101 : Why was consensus chosen instead of average. Also what was considered discordant, and what percentage of results were changed per discordance.

Line 100 : experienced ( not experiences)

Line 108 cox regression ( not cox's).    Also , typo. "statistically" , 

Line 123-126-133-135 etc : multiple error messages , please clean up.

Lines 124-135, please provide correlation coefficients alongwith p values.  Also please ensure if bonferroni corrections made for multiple correlations

Table 1 : better to report as only descriptive data without p values for correlations. 

Table 2 : Again clarify if multiple comparisons adjustments were made

The article is statistically very dense, with multiple exploratory analyses being conducted, the description of univerate analysis ( line 192-210) , including tables should be succinctly summarized in the body of manuscript and all tables should be moved to the appendix.  this will help the reader focus on the important information

Line 239 : please clarify that the benefit is only in intermediate-poor risk RCC

Line 266:  the biology of good risk and poor risk RCC is different, good risk patients tend to have a higher angiogenesis signature while poor risk patients have a higher T-cell effector signature etc - (please refer mcdermott et al 2018 nature medicine). Although most patients are localized in this study - on recurrence in the era of this study , all patients were likely treated with TKIs only, leading to a possible explanation of this difference, this should be captured in the discussion.

Line 293 : this sentence tries to capture the above, please explain this in better detail here. 

Final comment : The article must address associated with data mining in the abstracts and discussion section as multiple comparisons are made and outcomes should be classified as exploratory

Authors are to be commended on completion of a challenging and valuable study.

Author Response

Comments and Suggestions for Authors

The authors are to be commended on performing a very large study and examining a very important and novel question. The answers to these questions ( prognostic role of ICI in non CI treated/localized disease patients) , would be very informative in the design and conduct of potential adjuvant studies etc.

However, there are several issues that must be addressed. These include grammar to addressing/clarifying methodology. Some of the suggestions are enumerated below.

Line 16 , CSS , OS was analyzed "in a population not exposed to Immune checkpoint therapy" ( should be added )

Comment: We added the clarification.

Line 38 , should attempt to get more updated data than Heng et al's updat eform 2008 , VEGF TKIs and combination therapy is extending survival.

Comment: We added a recent analysis of survival in metastatic renal cell carcinoma. Survival here is 18 months.

Line 100 -101 : Why was consensus chosen instead of average. Also what was considered discordant, and what percentage of results were changed per discordance.

Comment: In less than 10% of the samples the pathologists did not reach the same value in the independent evaluation. The sample was then re-evaluated by both pathologists and a consensus was reached for all samples. Since for PD-1 and PD-L1 staining no staining was compared to positive staining with different extent and intensity, the calculation of mean values is not suitable for evaluation.

Line 100 : experienced ( not experiences)

Comment: Spelling corrected.

Line 108 cox regression ( not cox's).    Also , typo. "statistically" , 

Comment: Spelling corrected in the whole text.

Line 123-126-133-135 etc : multiple error messages , please clean up.

Comment: References to tables corrected, accordingly.

Lines 124-135, please provide correlation coefficients along with p values.  Also please ensure if bonferroni corrections made for multiple correlations

Comment: Correlation coefficients are added. The significance level was assumed to be 0.05. The significance level is listed in the captions of the tables. Bonferroni corrections results in a significance level of 0.05/48=0.00104. We have supplemented this accordingly in the caption to the table.

Table 1 : better to report as only descriptive data without p values for correlations.

Comment: Thank you for pointing that out. We included the p-values to facilitate visualization of statistical anomalies, as we believe this increases the readability of the table. We added a note on the Bonferroni correction in the caption of the table.

Table 2 : Again clarify if multiple comparisons adjustments were made

Comment: The significance level was assumed to be 0.05 as indicated. The Bonferroni correction results with a significance level of 0.05/25=0.002. We supplemented this accordingly in the caption of the table.

The article is statistically very dense, with multiple exploratory analyses being conducted, the description of univerate analysis ( line 192-210) , including tables should be succinctly summarized in the body of manuscript and all tables should be moved to the appendix.  this will help the reader focus on the important information

Comment: Thank you very much for your comment. For ease of reading, the univariate analysis tables have been moved to the Appendix. The results are referenced in the text.

Line 239 : please clarify that the benefit is only in intermediate-poor risk RCC

Comment: Clarification added.

Line 266:  the biology of good risk and poor risk RCC is different, good risk patients tend to have a higher angiogenesis signature while poor risk patients have a higher T-cell effector signature etc - (please refer mcdermott et al 2018 nature medicine). Although most patients are localized in this study - on recurrence in the era of this study , all patients were likely treated with TKIs only, leading to a possible explanation of this difference, this should be captured in the discussion.

Comment: Thank you very much for the addition. We referred to the results of the listed work and considered them accordingly in our discussion. We would also like to point out that we have already referred to the IMmotion010 study on adjuvant therapy with atezolizumab (NCT03024996), although no results have yet been published. We have also referred to the biomarker arms of the current studies on adjuvant therapy in renal cell carcinoma. Here, too, to our knowledge no results have been published so far.

Line 293 : this sentence tries to capture the above, please explain this in better detail here. 

Comment: Recent molecular characterization results have been referenced as indicated above.

Final comment : The article must address associated with data mining in the abstracts and discussion section as multiple comparisons are made and outcomes should be classified as exploratory

Comment: Survival outcomes in our study are not an explorative endpoint as we performed a retrospective study with a corresponding study design. The results correspond to expectations based on current literature.  The results on survival show a statistical correlation that could be reproduced in several statistical tests including multivariate cox analysis and thus are considered to be robust. Nevertheless, a validation of the results in an independent TMA would be an important adjunct in the future. We have supplemented this in the discussion.

Authors are to be commended on completion of a challenging and valuable study.

Comment: Thank you very much for the valuable review.

Reviewer 2 Report

In summary, this paper seeks to study the expression of PD-1, PD-L1, and CTLA-4 in the tumor microenvironment of patients with renal cell carcinoma (all histologies).  The novelty of this paper is trying to understand if CTLA-4 staining correlates with the biology and clinical prognosis of patients as well as the authors point out to reevaluate previously conflicting data regarding PD-1 staining in the tumor immune microenvironment of RCC. The authors have 342 tumor specimens in a TMA as an excellent resource upon which to do staining.  These 342 tumor specimens include all histologies, as well as both primary kidney and metastatic disease (primary and secondary not clearly defined in the paper).  I think the results are intriguing but several concerns in the methods could substantially be improved upon. CTLA-4 antibody used authors don't mention prior validation and do not perform staining with positive/ negative controls of TMA in other tissue or over expressing the protein.  The paper referenced in NSCLC looked at CTLA-4 using different antibodies for IHC.

 The choice of >2% positivity for CTLA-4 staining on tumor cells was based on a paper that looking at TIMC being approx 1% positive. Why did the authors choose tumor vs TIMC?  The staining in the IHC looks quite strong in the representative picture but found that many conclusions was drawn from a sample size of 20 patients out of 342 being considered positive.  It would have been nice to see in the results categories of CTLA-4 positive, e.g. >10%, >40%, >80% as other papers have done to characterize their antibodies and biology of the tumor.  

In results section authors suggest correlation with OS and CSS based on their TMA which could be made stronger if confirmed in a validation TMA. 

In discussion the authors make a statement see below regarding localized renal cell and the prognostic value of PD-1, PD-L1, and CTLA-4 but this doesn't seem to align with what the authors were testing in a TMA with both primary and metastatic disease.

"Although checkpoint inhibitors targeting these pathways are changing 239 therapy in metastatic renal cell carcinoma [11,35], the prognostic value of PD-1, PD-L1, and CTLA-4 240 expression in localized renal cell carcinoma remains still unclear."

Author Response

Comments and Suggestions for Authors

In summary, this paper seeks to study the expression of PD-1, PD-L1, and CTLA-4 in the tumor microenvironment of patients with renal cell carcinoma (all histologies).  The novelty of this paper is trying to understand if CTLA-4 staining correlates with the biology and clinical prognosis of patients as well as the authors point out to reevaluate previously conflicting data regarding PD-1 staining in the tumor immune microenvironment of RCC. The authors have 342 tumor specimens in a TMA as an excellent resource upon which to do staining.  These 342 tumor specimens include all histologies, as well as both primary kidney and metastatic disease (primary and secondary not clearly defined in the paper).

Comment: Thank you for your comment. All tumors investigated are primary manifestations of renal cell carcinomas.  Renal metastases from renal cell carcinomas and from tumors of other origin were not included in the study. Primarily metastasized tumors describe tumors that are already metastasized at diagnosis. Secondary metastasized tumors describe tumors that developed metastases during follow-up. We have added this in the Experimental Section. 

I think the results are intriguing but several concerns in the methods could substantially be improved upon. CTLA-4 antibody used authors don't mention prior validation and do not perform staining with positive/ negative controls of TMA in other tissue or over expressing the protein.  The paper referenced in NSCLC looked at CTLA-4 using different antibodies for IHC.

Comment: Before using the antibody reference in this study we tested several different commercially available CTLA4 antibodies from the literature and from different providers. Referring to the respective datasheets, these antibodies should have been suitable for usage on formalin-fixed and paraffin embedded tissues. None of them stained specifically in our tests. Finally, we tried BSB-88, which was not cited before. Like any other antibody, we initially tested the performance of BSB-88I on a TMA that contained a variety of normal tissues expected to be negative or positive for CTLA4, i.e. colon, breast, small intestine, adrenal, uterus, prostate, ovary, liver, tonsil, salivary gland, brain, heart, renal, appendix, skin, nerve, lung, testis, placenta, spleen, pancreas, endometrium, stomach and parotis tissue. Most tissues were negative, except for lymphatic tissue. We have therefore used lymphatic tissue in normal tonsil as a positive control. While evaluating CTLA4-staining on the RCC-TMA sections, negative control slides without the addition of primary antibody were included for each staining experiment.

The choice of >2% positivity for CTLA-4 staining on tumor cells was based on a paper that looking at TIMC being approx 1% positive. Why did the authors choose tumor vs TIMC?

Comment: Many thanks for the valuable comment. As in the publication mentioned, we also determined the proportion of CTLA-4 expressing cells of TIMC in all compartments of the tumor. In fact, this was misrepresented in the manuscript. We have adjusted the corresponding parts to avoid confusion.

The staining in the IHC looks quite strong in the representative picture but found that many conclusions was drawn from a sample size of 20 patients out of 342 being considered positive.  It would have been nice to see in the results categories of CTLA-4 positive, e.g. >10%, >40%, >80% as other papers have done to characterize their antibodies and biology of the tumor.

Comment: More than 10% pos. cells could only be shown in 2 cases, >40% pos. cells did not occur. Thus, we have omitted a statistical evaluation of these few cases.

In results section authors suggest correlation with OS and CSS based on their TMA which could be made stronger if confirmed in a validation TMA. 

Comment: Thank you for pointing that out. In fact, the results of this study can be further validated with the help of an independent TMA. We have added this remark to the discussion.

In discussion the authors make a statement see below regarding localized renal cell and the prognostic value of PD-1, PD-L1, and CTLA-4 but this doesn't seem to align with what the authors were testing in a TMA with both primary and metastatic disease.

"Although checkpoint inhibitors targeting these pathways are changing 239 therapy in metastatic renal cell carcinoma [11,35], the prognostic value of PD-1, PD-L1, and CTLA-4 240 expression in localized renal cell carcinoma remains still unclear."

Comment: The prognostic value of PD-1 and CTLA-4 is found in the entire cohort but also in the subgroup analysis with patients with no primary metastases (see Table 5, Table 8 and Table A2).

Reviewer 3 Report

This is a nicely conducted translational study.

The methods are solid and the results consistent with previous findings.

I suggest to include Involved kidney (left versus right)  in the multivariate analysis (see Prognostic factors associated with long-term survival in previously untreated metastatic renal cell carcinoma - https://academic.oup.com/annonc/article/18/2/249/250100)

Author Response

Comments and Suggestions for Authors

This is a nicely conducted translational study.

The methods are solid and the results consistent with previous findings.

I suggest to include Involved kidney (left versus right)  in the multivariate analysis (see Prognostic factors associated with long-term survival in previously untreated metastatic renal cell carcinoma - https://academic.oup.com/annonc/article/18/2/249/250100)

Comment: Thank you very much for the addition. We have calculated the influence of the localization (left vs. right) in our evaluation, but for OS (p=0.165) and CSS (p=0.10) we could not show a statistically significant connection (data not shown). Therefore, we did not consider localization in our multivariate analysis. However, we have implemented this in the discussion.